# Gender Differences among Nurses in Managing Arterial Puncture-Related Pain: A Multicenter Cross-Sectional Study

**DOI:** 10.3390/healthcare12050531

**Published:** 2024-02-23

**Authors:** Julio Alberto Mateos-Arroyo, Ignacio Zaragoza-García, Rubén Sánchez-Gómez, Paloma Posada-Moreno, Sara García-Almazán, Ismael Ortuño-Soriano

**Affiliations:** 1Department of Pneumology, Hospital General Nuestra Señora del Prado, 45600 Talavera de la Reina, Spain; juliomat@ucm.es; 2Department of Nursing, Faculty of Nursing, Physiotherapy and Podology, University Complutense of Madrid, 28040 Madrid, Spain; rusanc02@ucm.es (R.S.-G.); gerepa@ucm.es (P.P.-M.); iortunos@ucm.es (I.O.-S.); 3Instituto de Investigación Sanitaria Hospital 12 de Octubre (Imas12), InveCuid Group, 28041 Madrid, Spain; 4Fundación para la Investigación Biomédica del Hospital Clínico San Carlos (FIBHCSC), Instituto de Investigación Sanitaria Hospital Clínico San Carlos (IdISSC), 28040 Madrid, Spain; 5Intensive Care Unit, Hospital General Nuestra Señora del Prado, 45600 Talavera de la Reina, Spain; sgarciaa@sescam.jccm.es

**Keywords:** arterial puncture, gender differences, nurses, pain management

## Abstract

There is evidence that healthcare can be executed differentially depending on the gender of patients, researchers, and clinicians. The aim was to analyze the possible existence of nursing gender differences in pain management produced by arterial puncture for blood gas analysis. A cross-sectional, multicenter study designed was conducted in Castilla-la Mancha (Spain). Variables of interest were collected from nurses in the public health system of a European region through a questionnaire. Data were collected for four months; the primary outcome was the use of any intervention to reduce pain and the explanatory variable was the nurse’s gender. Bivariate analysis was carried out to assess associations between gender and pain-reducing interventions and a multivariate model was created with those factors that were relevant using logistic regression. A significantly higher proportion of men reported using some form of intervention (45% vs. 30%) and had more specific training (45.9% vs. 32.4%). The adjusted probability of using pain-reducing interventions by men was 71% higher than women. Thus, we found gender differences in the management of pain caused by arterial punctures performed by nurses as the main healthcare providers.

## 1. Introduction

The gas analysis of arterial blood samples represents a fundamental part of the diagnosis and treatment of critically ill respiratory patients [1]. The arterial blood sample can be obtained uniquely by direct arterial puncture or through a previously inserted catheter or arterial line [2]. The prevalence of this procedure increased significantly during the COVID-19 pandemic, as did the potential adverse events associated with it. This arterial puncture is unpleasant and painful for patients [3]. Subcutaneous infiltration of an anesthetic (e.g., lidocaine, mepivacaine) prior to direct arterial puncture significantly reduces the pain caused by arterial puncture [4,5,6]. It is paradoxical that, despite the existence of this method and other non-pharmacological methods proven to reduce pain, its use is not widespread in clinical settings [3,7,8].

In recent years, there has been a development of personalized medicine, and this must include gender medicine, which is considered the first step to being able to implement personalized medicine [9]. Sex-specific biological differences are documented in different health fields such as physiology, physiopathology, clinical manifestation, natural history, incidence, prevalence, treatment response, and mortality rates of key diseases [10]. However, not all aspects of health are related to biological sex, but gender, which is understood as a set of socio-cultural and political aspects, can modify or influence the health status of individuals [11]. This behavior also applies to the health of workers, which is the objective of occupational medicine. Different authors focus on gender differences in order to understand the health of workers not only at the biological level, but also at the psychological and social levels [9,12]. A recent review of the literature by Santoro et al. [9] describes a greater predisposition to infection at the occupational level by men in the physical sphere; on the other hand, at the psychosocial level, women present a greater likelihood of suffering stress, depression, and anxiety at work. Sorrentino et al. link it to occupational segregation as well as to increased domestic responsibilities attributed to women [13].

Likewise, Biswas et al. in their review reveal the existence of differences in occupational health and gender. In addition, they found the existence of social constructs about what is appropriate work for men and specific jobs or tasks for women. However, scientific evidence shows that these activities can be performed by both sexes without any problems. On the other hand, they show that women’s occupational hazards, such as bullying or harassment, are often less viable and less recognized than those of men [12].

These differences appear to affect not only the health of workers but there is also evidence that health care can be executed differentially depending on the gender of patients, researchers, and clinicians [14]. These differences can be understood from the concept of what is known as gender bias [15]. Specifically, this concept refers to the difference in the treatment of men and women with the same clinical diagnosis, which may have positive, negative or neutral consequences for their health [16]. In the country where this study was carried out, an arterial puncture is one of the techniques mainly performed by nurses both in our context [17], as in many other countries [18,19], and therefore the healthcare professionals who perform pain management. In addition, most registered nurses in our setting are female [20].

Pain experimentally induced by health professionals has been analyzed from a gender perspective. Robinson and Wise [21] designed an experiment in which they showed participants’ videos of volunteers experiencing varying degrees of pain produced by a cold pressor that reached temperatures between 1–3 °C. Among their findings was that women rated the observed pain 8–10 points higher than men on a 100-point scale. We also know, from a systematic review, that the contribution of race and gender by healthcare providers in general pain management seems limited, with contradictory results [22]. However, in this review, only one specifically analyzes nurses as healthcare providers, finding no significant differences in the approach to pain according to the gender of the professionals [23]. Furthermore, we do not know whether these differences could also exist in the pain produced by arterial punctures performed by nurses. Given the background described above, we hypothesize that there may be differences in nurses’ perception and management of pain produced by arterial puncture for blood gas analysis. Universal access to pain treatment, as a guarantor of a correct state of health, is considered a human right [24]; hence, the reduction of pain induced by medical procedures should be a priority in the clinical setting. In general, the causes of underuse of analgesic methods should be investigated. The importance of this study lies in the fact that, if demonstrated, the existence of possible gender differences in nurses may focus efforts to mitigate unjustified distinctions in pain management. As a result, specific training could be designed in the future with the aim of eradicating possible stereotypes. Therefore, the aim of our study is to analyze the possible existence of gender differences in pain management produced by arterial puncture performed by nurses as the main healthcare providers.

## 2. Materials and Methods

### 2.1. Study Design

A cross-sectional, multicenter study was designed to achieve the main objective. The STROBE statement was followed to report the results of this study [25].

### 2.2. Study Setting and Sampling

#### 2.2.1. Setting

This research used an ad hoc questionnaire to collect differential factors between genders in a European region, Castilla-La Mancha (Spain), with approximately 10,000 nurses. The survey was available for 4 months (August–November 2022) to be completed by the nurses. The survey was available to the entire population of nurses who were the subject of this study via corporate mail.

#### 2.2.2. Participants

This study included registered nurses who were currently working in the public health system, who had ever performed an arterial blood gas analysis (ABG) puncture, and who agreed to complete the survey designed for data collection and thus participate in the study. We excluded undergraduates or postgraduates with a training contract and other healthcare professionals who could perform ABG puncture, such as specialist physicians (e.g., anesthesiology, Intensive Care Medicine).

#### 2.2.3. Sample Size

In the absence of similar studies, a general estimate of the prevalence of the use of local anesthetic injections to reduce pain during ABG puncture was used to calculate the sample size. According to the study conducted by Ballesteros-Peña et al. [7], pain interventions had an employment rate of 23% at a 95% confidence level with an accuracy of 3.50 units and an expected loss of 1%. Therefore, a minimum of 528 participants is recommended.

### 2.3. Data Collection and Analysis

#### 2.3.1. Variables

The primary outcome was the use of any intervention to reduce pain from ABG puncture. This variable was dichotomized (yes/no). The main explanatory variable was gender, initially establishing three categories: male, female, and non-binary. Finally, this variable was dichotomous, since no response was obtained from any non-binary nurse. Other variables were collected: sociodemographic variables (age, work experience as a nurse, healthcare workplace, hospital department, and employment relationship); ABG puncture-related variables (puncture site, number of ABG punctures per month, perception of pain generated, specific ABG puncture training, self-skill perception, and Allen test frequency); individual perception of interventions efficacy (ultrasound-guided puncture technique, use of fine gauge needles, administration of topical anesthetic creams, local anesthetic infiltration, use of cold sprays, and application of ice); and variables related to local anesthetic infiltration as a gold standard in pain reduction [26] (frequency of use, reasons for non-use, and desire to be self-administered).

#### 2.3.2. Data Sources/Measurement

Data were collected using an ad hoc questionnaire based on other publications on knowledge and attitudes about ABG puncture [7,8]. This survey consisted of 17 questions. In addition, 10 preliminary questions were asked about their sociodemographic characteristics. Most of the questions were closed-ended and single-answer questions. All quantitative variables that measured frequencies used a 5-point Likert scale. Age and nurse experience were collected in years and the validated instrument used to measure the perception of pain generated was the numerical rating scale (NRS) [27].

#### 2.3.3. Bias

This type of cross-sectional survey usually has a low response rate [28]. It should be noted that the survey method used may lead to information bias because there may be important differences between those who responded and those who did not. To minimize this possible bias, we have tried to choose a sufficiently large and representative sample of the population studied. To avoid response bias, several reminders were sent, and, finally, a call for participation was made through social networks, so that nurses who initially did not want to participate might eventually be encouraged to participate.

We have added this issue to the bias section.

#### 2.3.4. Statistical Methods

Categorical variables were expressed as frequency and percentage. The Chi-squared (χ^2^) or Fisher’s test was used for between-group comparisons. The normality of quantitative variables was tested using the Kolmogorov–Smirnov test. Quantitative variables were expressed as the median and interquartile range (Q1–Q3) for non-normal distributions, and groups were compared using the Mann–Whitney test as appropriate. Odds ratios (OR) with their respective confidence intervals were also obtained for relevant variables when possible (2 × 2). These measures of association were adjusted using the Mantel–Haenszel test to assess the presence of confounding. Gender was adjusted by the factors that were relevant in the bivariate analysis using a multivariate logistic regression analysis to explain the use of pain-reducing interventions. The variables were entered separately together with the gender and the models were compared using Akaike’s information criterion (AIC). The Hosmer–Lemeshow test was used to assess the adequacy of the final model fit. In addition, we calculated the area under the ROC curve with its respective confidence intervals to check the discriminative power of this model. Observations from which all variables could not be obtained were considered to be missing data. For all comparisons, a statistical significance level of *p* < 0.050 was established. However, due to the multiple comparisons made with the different demographic and practice-related factors, all *p*-values were evaluated using the Benjamini–Hochberg procedure to control for false discovery rates. Data were analyzed using the Jamovi statistical open-source package based on R software version 2.2.5.0 [29].

### 2.4. Ethical Considerations

This study and the informed consent obtained approval from the ethics research committee of a public third-level hospital with code 12/2022. This study complied with current legislation and was in accordance with the Declaration of Helsinki. All subjects agreed to participate in this study prior to enrollment.

## 3. Results

### 3.1. Participants

A total of 584 questionnaires were received, of which 528 were finally validated. The reasons for exclusion were never having performed an ABG puncture (n = 43), not currently working as a nurse (n = 8), not working in the region studied (n = 3), and not giving consent (n = 2).

### 3.2. Descriptive Data

Most of the surveyed nurses were women (79%), had a permanent employment contract (43.9%), and worked in hospitals (62.9%). The departments with the highest response rate were the emergency department (20.5%), followed by medical units (18.4%) and the ICU (14.8%). Regarding ABG puncture, 35.2% have specific training and 92% choose the radial artery preferentially, reporting a median skill in performing the technique of 4 points out of 5 and a median in the perception of pain generated with the NRS of 7 out of 10. In addition, the three pain-reducing interventions that most nurses perceived as effective were the use of topical anesthetic creams (55.3%), cryotherapy using cold sprays (41.9%), and ultrasound-guided puncture technique (40.9%). The proportion of nurses who reported never using local anesthetic infiltration, which is considered to be the gold standard, was 83.1%. On the other hand, 85.4% stated that they would be willing to receive any type of pain-reducing intervention if they were the ones to receive an ABG puncture.

### 3.3. Primary Outcome and Gender

The main characteristics of the participants according to gender are shown in Table 1. In total, 66.5% of the nurses reported not using any pain-reducing strategies for ABG puncture.

The proportion of nurses who used any pain intervention was higher in men (45%) than in women (30%) with an OR = 1.87 [95% CI: 1.22–2.87]; *p* = 0.004. There were no significant differences between genders in the main sociodemographic variables: age, experience, place of work, and length of contract.

As shown in Table 2, there were statistically significant differences between men and women in specific training, and perception of pain generated (Appendix A). Furthermore, in Table 3, it can be seen that there are differences in terms of perception of efficacy of local anesthetic infiltration and frequency of use of anesthetics. However, no significant differences were found between male and female nurses in the perceived efficacy of other methods to reduce pain caused by ABG puncture (e.g., use of topical anesthetic creams or cryotherapy). There are also no significant differences in the gender of the nurses when it comes to requiring any type of intervention in the hypothetical case of being the one to receive an ABG puncture.

There were also differences in two of the reasons for not using the gold standard: lack of knowledge in infiltration and lack of protocols in the work units (Table 3). These gender differences were maintained in the subgroups according to the perception of the pain generation, and partially of the use of pain-reducing interventions in the specific training variable (Table 2), although in both cases these differences are lost with the Benjamini-Hochberg correction procedure. Partial differences were also maintained regarding the use of pain interventions in the lack of protocols or instructions in the unit (Table 3). The measure of the unadjusted association between gender and the variables described above can be seen in Table 4. In general, male nurses had higher odds of having specific training in ABG puncture (78%), higher odds on the perception of the efficacy of infiltrated anesthetics (66%), and higher odds on the frequency of use of the gold standard. This probability in the nurses’ gender is 2.43 times higher among those who rarely use it with respect to those who never infiltrate anesthetics and increases gradually reaching ×19.41 times among those who always use this anesthetic method. On the other hand, male nurses had lower odds of a lack of knowledge of the technique of anesthetic injection (49%) and lower odds of working in units without instructions or protocols (43%) than female nurses. Finally, the association between gender and primary outcome was adjusted for other covariates (Table 5). According to this model, the probability of using pain-reducing interventions by men is 71% higher than women. Regarding the validity of the model, the Hosmer–Lemeshow goodness-of-fit test obtained a *p*-value of 0.758, indicating that the model is valid between observed and predicted. Furthermore, the discriminative ability of the model for the dependent variable use of pain-relieving interventions was significant in the area under the ROC curve (AUCROC = 0.603 CI95% (0.551–0.654); *p* < 0.001) (Appendix A).

## 4. Discussion

We found differences in ABG puncture pain management in terms of nurse gender as a care provider. The existence of these gender differences may be explained by some of the factors analyzed in this study.

### 4.1. Gender Proportion

It should be noted that the proportion of female nurses in the selected sample was significantly higher (79%). This amount is close to that published by the National Institute of Statistics [20] of the country where this study was conducted for the preceding year (83.5%). This statement is already in line with the data published by KFF in the USA, which indicate that the number of female nurses has remained stable for many years. Therefore, it seems that the selected sample could be representative of the study population in terms of gender.

### 4.2. Differential Factors between Genders

The main finding of this study is that a greater proportion of male nurses than female nurses use some strategy to minimize ABG puncture pain (45.0% vs. 30.0%). If we also focus exclusively on local anesthetic infiltration, which is recommended by WHO [2] and considered to be the most effective method [30], in our study, a higher proportion of men perceive it as effective (39.6% vs. 28.3%) and a lower proportion of men never use it (67.6% vs. 87.3%). Although we did not find similar studies evaluating the use of pain minimization strategies by gender, different authors [31] also found in other contexts that healthcare professionals do not use local anesthetic infiltration to reduce pain when performing ABG punctures. Nevertheless, as in our study, Zinchenko et al. found that most participants perceived ABG as “quite painful” (61%) or “extremely painful” (20%). On the other hand, a recent study in a hospital emergency department [32] observed low rates of local anesthetic use for ABG puncture, again with lower rates of use by female nurses (14.1% vs. 22.2%).

However, these rates of utilization of pain-reducing interventions by gender contrast with the perception of pain generated by ABG puncture. The greater intensity of pain perceived by female nurses found in our study is in line with what has been published in other experimental studies. Robinson and Wise [21], using videos of healthy volunteers subjected to pain, determined that it was women who, watching the same video with participants of the same sex, rated the pain observed as more intense than men.

### 4.3. Inequality in ABG Puncture Training

Gender is shown in this study as a differential factor in the specific training, with a higher proportion of male nurses with ABG puncture formation than female nurses (45.9% vs. 32.4%). In addition, local infiltration of anesthetics is higher among men than among women (39.6% vs. 28.3%). We believe that this could be due to a higher education or degree of specialization of nurses in pain management. Despite not being able to confirm this fact, in our study or other works, most studies comparing educational attainment by gender report that men have higher degrees and therefore more education than women. Greene et al. [33], who studied the salary gap for nurse practitioners in the United States, found that 3.3% of men had doctorates compared to 1.5% of women. In addition, they note that knowledge provides men with greater autonomy and higher income levels. Similarly, their study of retirement and gender in nursing found that 9.4% of men are doctors compared to 4.4% of women.

This theory of inequality in ABG puncture training by gender is also reinforced by the higher proportion of female nurses reporting a lack of knowledge of subcutaneous anesthetic infiltration technique (18.9% vs. 31.2%). These disparities in training, knowledge, and perceptions necessarily result in a different use of local anesthetics as well. The proportion of male nurses who never use local anesthetic infiltration is considerably lower than that of female nurses (67.6% vs. 87.3%) and, on the contrary, higher in those who always use this pain-reducing intervention (7.2% vs. 0.5%). It is important to consider that various authors have noted discrepancies in the professional conduct of male and female nurses. According to Torkelson [34], male nurses tend to prioritize decision making and responsibility acquisition by obtaining knowledge and skills, with the aim of impacting the quality of patient care and achieving professional recognition versus care, vigilance, and safety, without giving as much importance to the technical competence shown by female nurses, affirming the existence of the stereotype of the male nurse who seeks independence versus the protection of the female nurse.

Clavero et al. [35] conducted a study on the influence of gender on nursing practices and found that female nurses perceive male nurses as more practical and focused on efficient care delivery, while male nurses perceive female nurses as more focused on detail and aesthetic aspects. However, no differences are found in terms of the implementation of technical aspects. This suggests that, despite differences in the application of care by both sexes, the problem may be more focused on a lack of knowledge and motivation for its application.

Regarding training inequality, although the formative programs of the different faculties at the national level are regulated by a standard, there could be differences in terms of the development of autonomy and self-confidence of each student, as described by Meyerson et al. [36] in their research with surgery residents, where women, despite demonstrating clinical performance the same as men, they express having less autonomy and underestimate their clinical performance. Despite this, we believe that the most significant difference lies in access to specific postgraduate training, which is not regulated in our context. Another possible explanation may be that female health workers have less time away from the workplace for specific training. Mele et al. [37], during the COVID-19 pandemic, found that women working in health systems reported that domestic duties increased considerably with respect to men in the same situation.

Gender equality is a controversial and topical issue in our society. The World Health Organization recognizes that this is a key aspect of the living conditions of women and men [38]. Gender inequality is associated with several negative outcomes, including differences in education. This aspect is very evident and measurable in low- and middle-income countries and less visible in high-income countries, as is the case in our context [39]. This is important because, according to Milner et al., gender differences may be masked in high-income countries because they are not as obvious [39]. This is interesting considering our results because, although economically both men and women have access to higher education, it is possible that family burdens and social determinism may lead to women having less postgraduate education than men. Although there is talk of the gender gap in nursing [40], further studies are needed to find explanatory factors for these gender differences related to access to specialized training.

### 4.4. Generalizability

We believe that the findings of this study could be generalized to other nationwide nursing populations in which this study was conducted. However, due to the likely heterogeneity of the different postgraduate training programs, future studies in other nursing population contexts are needed. To the best of our knowledge, there is no research that jointly addresses gender and all types of pain-reducing ABG puncture interventions (including nonpharmacologic). However, we have found references from other authors, in our context, that confirm the low rates of use of local anesthetics [7,8] and the lower proportion of use of local anesthetics by women [32]. The results of these future studies may prove crucial in understanding how different training programs and different competencies acquired by nurses may impact the management of pain induced by arterial puncture for blood gases.

## 5. Limitations

According to the existing scientific literature, it is to be expected that pain management will be different depending on the sex of the patient undergoing ABG puncture [14,21,41]. Our study did not consider this factor, which could limit the results, particularly in terms of differing pain perception. On the other hand, the number of female nurses is overrepresented in this study. However, these data are in line with the figures provided by the WHO [42], in which women represent the main healthcare workforce. Moreover, as previously considered, this sample responds to the population reality of the geographic area studied. It is crucial to note the study’s multicenter aspect, which guarantees representative participation from nurses.

## 6. Implications for Practice

Our study has identified significant differences in pain management between male and female nurses, highlighting the need for targeted training strategies to address this disparity. Additionally, further analysis is required to understand the reasons behind the differences in ABG and pain management training between genders. It is imperative that health policies are reformed to combat gender inequality in the nursing profession.

Addressing gender differences is a multifaceted issue that requires not only workplace intervention but also political support to effect social change. Our study highlights the importance of effective nursing management to identify and address worker limitations. For instance, training programs should be tailored to facilitate work–life balance. New technologies may aid in this regard, as they enable professionals to receive training at their convenience. Additionally, it is crucial to review work protocols, including the use of analgesic measures during the arterial puncture technique.

## 7. Conclusions

We discovered gender discrepancies in pain management resulting from arterial punctures carried out by male and female nurses, who are the primary healthcare providers. A correlation between gender and specific training in this area was observed, which appears to contribute to the low perceived efficacy and subsequent underuse of pain-reducing methods by female nurses. This study sheds light on the potential for nursing managers to provide appropriate motivation and targeted ABG training for male and female patients, ultimately resulting in reduced pain levels during ABG procedures.

## Figures and Tables

**Table 1 healthcare-12-00531-t001:** Characteristics of nurses and comparison of sociodemographic and laboral variables between males and females.

	Total (n = 528)				Use of Pain Interventions: YES (n = 177)				Use of Pain Interventions: NO (n = 351)			
	Male (n = 111)	Female (n = 417)	*p*-Value	B-H *p*-Value	Male (n = 50)	Female (n = 127)	*p*-Value	B-H *p*-Value	Male (n = 61)	Female (n = 290)	*p*-Value	B-H *p*-Value
Age in years ^†^	41 [32–49]	42 [33–49]	0.883	0.942	41 [33–47]	43 [35–49]	0.679	0.933	41 [32–50]	41 [32–49]	0.734	0.897
Work experience as nurse in years ^†^	18 [6–26]	17 [8–24]	0.942	0.942	41 [33–47]	43 [35–49]	0.773	0.944	15 [6–26]	17 [7–24]	0.731	0.897
Nurse’s workplace ^‡^			0.562	0.876			0.226	0.640			0.447	0.897
(a) Hospital	71 (64.0)	261 (62.6)	0.281	0.772	30 (60.0)	86 (67.7)	0.060	0.330	41 (67.2)	175 (60.3)	0.684	0.897
Pneumology	3 (4.3)	20 (7.6)			1 (3.6)	5 (5.6)			2 (4.8)	15 (8.7)		
Internal medicine	9 (12.9)	30 (11.5)			3 (10.7)	10 (11.2)			6 (14.3)	20 (11.6)		
Other medical units	9 (12.9)	52 (19.8)			2 (7.1)	15 (16.9)			7 (16.7)	37 (21.4)		
Surgical units	10 (14.3)	31 (11.8)			6 (21.4)	8 (9.0)			4 (9.5)	23 (13.3)		
Emergency	17 (24.3)	51 (19.5)			6 (21.4)	13 (14.6)			11 (26.2)	38 (22.0)		
ICU	11 (15.7)	38 (14.5)			4 (14.3)	24 (27.0)			7 (16.7)	14 (8.1)		
Intermediate care	1 (1.4)	3 (1.1)			1 (3.6)	1 (1.1)			0 (0)	2 (1.2)		
Clinical laboratory	7 (10.0)	10 (3.8)			4 (14.3)	2 (2.2)			3 (7.1)	8 (4.6)		
Outpatient service	3 (4.3)	27 (10.3)			1 (3.6)	11 (12.4)			2 (4.8)	16 (9.2)		
(b) Primary healthcare	32 (28.8)	137 (32.9)			14 (28.0)	36 (28.3)			18 (29.5)	101 (34.8)		
(c) Outpatient healthcare	5 (4.5)	12 (2.9)			5 (10.0)	4 (3.1)			0 (0)	8 (2.8)		
(d) Other	3 (2.7)	7 (1.7)			1 (2.0)	1 (0.8)			2 (3.3)	6 (2.1)		
Type of employment contract ^‡^			0.708	0.876			0.565	0.888			0.939	0.939
Permanent	50 (45.0)	182 (43.6)			23 (46.0)	56 (44.1)			27 (44.3)	126 (43.4)		
Interim	20 (18.0)	90 (21.6)			9 (18.0)	32 (25.2)			11 (18.0)	58 (20.0)		
Temporary	41 (36.9)	145 (34.8)			18 (36.0)	39 (30.7)			23 (37.7)	106 (36.6)		

Note. * Statistically significant values after adjustment with the Benjamini–Hochberg procedure. B-H: Benjamini–Hochberg adjustment. ^†^ Expressed as median [IR] and Mann–Whitney test. ^‡^ Expressed as n (%) and Chi-squared test or Fisher’s exact test.

**Table 2 healthcare-12-00531-t002:** ABG puncture-related variables between males and females.

	Total (n = 528)				Use of Pain Interventions: YES (n = 177)				Use of Pain Interventions: NO (n = 351)			
	Male (n = 111)	Female (n = 417)	*p*-Value	B-H *p*-Value	Male (n = 50)	Female (n = 127)	*p*-Value	B-H *p*-Value	Male (n = 61)	Female (n = 290)	*p*-Value	B-H *p*-Value
Perception of pain generated ^†^	7 [6–7]	7 [6–8]	**0.001 ***	**0.011 ***	7 [6–7]	7 [6–8]	**0.021**	0.231	7 [6–7]	7 [7–8]	**0.029**	0.160
Puncture site ^‡^			00.717	00.876			0.469	0.878			0.816	0.898
Radial	101 (91.0)	385 (92.3)			43 (86.0)	116 (91.3)			58 (95.1)	269 (92.8)		
Humeral	10 (9.0)	30 (7.2)			7 (14.0)	10 (7.9)			3 (4.9)	20 (6.9)		
Other	0 (0)	2 (0.5)			0 (0)	1 (0.8)			0 (0)	1 (0.3)		
Specific ABG puncture training ^‡^	51 (45.9)	135 (32.4)	**0.008 ***	0.044	23 (46.0)	51 (40.2)	0.479	0.878	28 (45.9)	84 (29.0)	**0.010**	0.110
ABG puncture/month ^‡^			0.632	0.876			0.967	0.967			0.402	0.897
0 to 5	79 (71.2)	313 (75.1)			36 (72.0)	94 (74.0)			43 (70.5)	219 (75.5)		
6 to 10	11 (9.9)	42 (10.1)			6 (12.0)	16 (12.6)			5 (8.2)	26 (9.0)		
11 to 15	9 (8.1)	21 (5.0)			2 (4.0)	5 (3.9)			7 (11.5)	16 (5.5)		
>15	12 (10.8)	41 (9.8)			6 (12.0)	12 (9.4)			6 (9.8)	29 (10.0)		
Skill perception performing arterial puncture ^†^	4 [3–4]	4 [3–4]	0.087	0.319	4 [3–5]	4 [3–4]	0.233	0.640	4 [3–4]	4 [3–4]	0.358	0.897
Allen test frequency ^‡^			0.714	0.876			0.928	0.967			0.588	897
Never	29 (26.1)	127 (30.5)			10 (20.0)	33 (26.0)			19 (31.1)	94 (32.4)		
Rarely	33 (29.7)	112 (26.9)			13 (26.0)	29 (22.8)			20 (32.8)	83 (28.6)		
Sometimes	26 (23.4)	97 (23.3)			13 (26.0)	34 (26.8)			13 (21.3)	63 (21.7)		
Often	12 (10.8)	32 (7.7)			6 (12.0)	13 (10.2)			6 (9.8)	19 (6.6)		
Always	11 (9.9)	49 (11.8)			8 (16.0)	18 (14.2)			3 (4.9)	31 (10.7)		

Note. * Statistically significant values in bold (*p* < 0.050); B-H: Benjamini–Hochberg adjustment. ^†^ Expressed as median [IR] and Mann–Whitney test. ^‡^ Expressed as n (%) and Chi-squared test or Fisher’s exact test.

**Table 3 healthcare-12-00531-t003:** Perceptions and use of pain-reducing interventions and comparison between males and females.

	Total (n = 528)			Use of Pain Interventions: YES (n = 177)			Use of Pain Interventions: NO (n = 351)		
	Male (n = 111)	Female (n = 417)	*p*-Value	Male (n = 111)	Female (n = 417)	*p*-Value	Male (n = 111)	Female (n = 417)	*p*-Value
Nurse perception of pain reduction from interventions									
Ultrasound-guided puncture technique	46 (41.4)	170 (40.8)	0.898	22 (44.0)	59 (46.5)	0.768	24 (39.3)	111 (38.3)	0.876
Use of fine gauge needles	33 (29.7)	96 (23.0)	0.144	18 (36.0)	39 (30.7)	0.498	15 (24.6)	57 (19.7)	0.386
Use of topical anesthetic creams	66 (59.5)	226 (54.2)	0.322	27 (54.0)	73 (57.5)	0.674	39 (63.9)	153 (52.8)	0.111
Local anesthetic infiltration	44 (39.6)	118 (28.3)	**0.021 ***	25 (50.0)	44 (34.6)	0.060	19 (31.1)	74 (25.5)	0.366
Cryotherapy (cold sprays)	46 (41.4)	175 (42.0)	0.921	16 (32.0)	59 (46.5)	0.080	30 (49.2)	116 (40.0)	0.186
Cryotherapy (ice)	34 (30.6)	142 (34.1)	0.497	13 (26.0)	44 (34.6)	0.268	21 (34.4)	98 (33.8)	0.924
Local anesthetic infiltration frequency			**<0.001 ****			**0.030 ***			**<0.001 ****
Never	75 (67.6)	364 (87.3)		27 (54.0)	90 (70.9)		48 (78.7)	274 (94.5)	
Rarely	10 (9.0)	20 (4.8)		5 (10.0)	10 (7.9)		5 (8.2)	10 (3.4)	
Sometimes	12 (10.8)	21 (5.0)		7 (14.0)	15 (11.8)		5 (8.2)	6 (2.1)	
Often	6 (5.4)	10 (2.4)		5 (10.0)	10 (7.9)		1 (1.6)	0 (0)	
Always	8 (7.2)	2 (0.5)		6 (12.0)	2 (1.6)		2 (3.3)	0 (0)	
Nurses’ reasons for not infiltrating local anesthetics									
Lack of knowledge of the anesthetic injection technique	21 (18.9)	130 (31.2)	**0.011 ***	6 (12.0)	25 (19.7)	0.227	15 (24.6)	105 (36.2)	0.083
Lack of protocols or instructions in my unit	52 (46.8)	254 (60.9)	**0.008 ***	26 (52.0)	59 (46.5)	0.508	26 (42.6)	195 (67.2)	**<0.001 ****
Two punctures when I usually puncture on the first attempt	36 (32.4)	119 (28.5)	0.423	14 (28.0)	45 (35.4)	0.345	22 (36.1)	74 (25.5)	0.093
Because it is not effective and, finally, the same pain is produced	6 (5.4)	14 (3.4)	0.315	1 (2.0)	4 (3.1)	0.678	5 (8.2)	10 (3.4)	0.096
Because more time is needed to administer the anesthesia and wait for it to take effect	37 (33.3)	102 (24.5)	0.059	21 (42.0)	35 (27.6)	0.063	16 (26.2)	67 (23.1)	0.601
A medical prescription is required	27 (24.3)	118 (28.3)	0.405	14 (28.0)	35 (27.6)	0.953	13 (21.3)	83 (28.6)	0.244
If a work partner had to perform an arterial puncture on you, would you like them to use some intervention to reduce pain?			0.143			0.517			0.280
No	16 (14.4)	61 (14.7)		6 (12.0)	17 (13.5)		10 (16.4)	44 (15.2)	
Yes, local anesthetics	37 (33.3)	101 (24.3)		17 (34.0)	32 (25.4)		20 (32.8)	69 (23.8)	
Yes, including non-pharmacological	58 (52.3)	254 (61.1)		27 (54.0)	77 (61.1)		31 (50.8)	177 (61.0)	

Note. * Statistically significant values in bold (*p* < 0.050), ** (*p* < 0.001). All variables expressed as n (%) and Chi-squared test or Fisher’s exact test.

**Table 4 healthcare-12-00531-t004:** Measures of association between significant variables in the bivariate analysis (gender: male vs. female).

				95% CI OR Exp(β)	
	Categories	*p*-Value	OR Exp(β)	Lower	Upper
Specific ABG puncture training	Yes vs. No	**0.008 ***	1.78	1.16	2.72
Local anesthetic infiltration efficacy perception	Yes vs. No	**0.021 ***	1.66	1.08	2.57
Local anesthetic infiltration frequency	Rarely vs. Never	**0.030 ***	2.43	1.09	5.39
	Sometimes vs. Never	**0.008 ***	2.77	1.31	5.88
	Often vs. Never	**0.044 ***	2.91	1.03	8.26
	Always vs. Never	**<0.001 ****	19.41	4.04	93.25
Lack of knowledge of the anesthetic injection technique	Yes vs. No	**0.011 ***	0.51	0.31	0.86
Lack of protocols or instructions in my unit	Yes vs. No	**0.008 ***	0.57	0.37	0.86

Note. * Statistically significant values in bold (*p* < 0.050), ** (*p* < 0.001); OR: odds ratio. Exp(β): exponential of beta; 95% CI: 95% confidence interval.

**Table 5 healthcare-12-00531-t005:** Multivariable explanatory model of the use of pain-reducing interventions where gender is adjusted for other factors.

				95% CI OR Exp(β)	
	Categories	*p*-Value	OR Exp(β)	Lower	Upper
Gender	Male vs. Female	**0.016 ***	1.71	1.10	2.64
Local anesthetic infiltration efficacy perception	Yes vs. No	**0.012 ***	1.65	1.12	2.43
Specific ABG puncture training	Yes vs. No	0.085	1.40	0.95	2.05

Note. * Statistically significant values in bold (*p* < 0.050); OR: odds ratio. Exp(β): exponential of beta; 95% CI: 95% confidence interval.

## Data Availability

The data presented in this study are available on request from the corresponding author.

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
