# Peer review of "Gender Differences among Nurses in Managing Arterial Puncture-Related Pain: A Multicenter Cross-Sectional Study"

_healthcare, 2024, doi:10.3390/healthcare12050531_

Round 1

Reviewer 1 Report (Previous Reviewer 1)

Comments and Suggestions for Authors

Dear authors,

Thank you for the opportunity to review your manuscript “Gender Differences Among Nurses in Managing Arterial Puncture-Related Pain: A Multicenter Cross-Sectional Study”; the study is very interesting and well thought out and the quality of the writing has been improved since the first version of this manuscript.

My only concern remains that the introduction section is really brief. You could expand on gender differences in occupational health (a couple of examples if you need some articles: Santoro et al “Occupational hazards and gender differences: a narrative review”, Biswas et al “Sex and Gender Differences in Occupational Hazard Exposures: a Scoping Review of the Recent Literature”).

All other suggested improvements have been carried out by the authors.

Author Response

Dear authors,

Thank you for the opportunity to review your manuscript “Gender Differences Among Nurses in Managing Arterial Puncture-Related Pain: A Multicenter Cross-Sectional Study”; the study is very interesting and well thought out and the quality of the writing has been improved since the first version of this manuscript.

My only concern remains that the introduction section is really brief. You could expand on gender differences in occupational health (a couple of examples if you need some articles: Santoro et al “Occupational hazards and gender differences: a narrative review”, Biswas et al “Sex and Gender Differences in Occupational Hazard Exposures: a Scoping Review of the Recent Literature”).

Thank you very much for your comments, as well as for the recommended bibliography, we believe it has been of great help to finish shaping the introduction. We have gone deeper into the topic of gender in occupational health. We believe that now the introduction makes more sense.

All other suggested improvements have been carried out by the authors.

We are grateful for their evaluation and also for their comments and contributions, which we believe have been of great help in improving the manuscript.

Reviewer 2 Report (New Reviewer)

Comments and Suggestions for Authors

This is an evidence-based study to identify differences in pain approach by gender of professionals during arterial puncture, and overall, we believe that the study followed the STROBE procedures closely.

Based on the results and discussion of this study, the following questions arise. We would be grateful if you could modify the introduction or discussion to reflect these:

1) In some countries, arterial puncture is not the primary role of nurses; the paper may seem to imply that arterial puncture is the primary role of nurses; therefore, further explanation of role differences between countries is needed.

2) Why are male nurses more likely to have higher degrees and more specific training in ABG puncture? It's hard to disagree with these results entirely, and I think it's worth mentioning the cultural differences between countries.

3) If female nurses report higher patient pain intensity, why do they use fewer pain reduction interventions? This dilemma deserves further discussion.

4) In conclusion, I am confused as to whether to view the results of this study as a gender difference or an education difference.

That's all for now. We wish you all the best in your revision work and look forward to publishing this manuscript as a research paper.

Yours sincerely, Reviewer

Author Response

This is an evidence-based study to identify differences in pain approach by gender of professionals during arterial puncture, and overall, we believe that the study followed the STROBE procedures closely.

Thank you very much for your kind words. We are delighted about it and we are ready to comment on your suggestions.

Based on the results and discussion of this study, the following questions arise. We would be grateful if you could modify the introduction or discussion to reflect these:

1) In some countries, arterial puncture is not the primary role of nurses; the paper may seem to imply that arterial puncture is the primary role of nurses; therefore, further explanation of role differences between countries is needed.

It is true that we are not aware of the regulation regarding the use of invasive techniques in all the countries of the world, especially arterial puncture, which is why we agree with the reviewer that we cannot state that it is the primary role of nurses. But on the other hand, in our context it is a primary role as well as in other contexts such as in Egypt or Anglo-Saxon countries. That is why we modify the wording to mean that it is a role mainly performed by nurses in our context as well as in other parts of the world. We show you some verbatim from some studies conducted in England and Egypt where nurses mainly perform this procedure.

“This has traditionally been the role of the doctor, however by using an education and training package along with a competency-based assessment, nurses can now perform this extended role”

  1. Dodds S., Williamson G. Nurse-led arterial blood gas sampling for patients. Nurs Times. 2007;103(8):44-5.

“Nurses are usually involved in taking and analyzing the ABGs and normally they report these results to the doctors or anesthesiologists”

  1. Mohammed HM., Abdelatief DA. Easy blood gas analysis: Implications for nursing. Egyptian Journal of Chest Diseases and Tuberculosis. 2016;65(1):369-76, doi: 10.1016/j.ejcdt.2015.11.009.

We have therefore modified the previous statement in the manuscript to read: “…arterial puncture is one of the techniques mainly performed by nurses both in our context as in many other countries.”

2) Why are male nurses more likely to have higher degrees and more specific training in ABG puncture? It's hard to disagree with these results entirely, and I think it's worth mentioning the cultural differences between countries.

Thank you very much for the suggestions, we think this aspect is very relevant today. We have reviewed the literature and added this aspect both in the introduction and in the discussion, addressing the issue of gender and possible inequalities.

3) If female nurses report higher patient pain intensity, why do they use fewer pain reduction interventions? This dilemma deserves further discussion.

We agree with your statement. This is a very interesting dilemma that should be addressed in future research. In our case, we have reviewed the scientific evidence and the only thing we can comment on in this regard is that some authors affirm that nurses in general tend to have a lower academic level and, in this particular case, less knowledge about pain management, both in its measurement and in the use of pharmacological and non-pharmacological interventions, in the knowledge gap. We believe that this may be the problem. In fact, our results show that women receive less training than men. This could mean that although female nurses perceive it as a painful technique, the barrier of lack of knowledge in its management leads to treatment failure.

Reference:

Albaqawi H., Maude P., Shawhan-Akl L. Saudi Arabian Nurses’ Knowledge and Attitudes Regarding Pain Management: Survey Results Using the KASRP. International Journal of Health Sciences & Research. 2016; 6 (12), 150–164. https://www.ijhsr.org/IJHSR_Vol.6_Issue.12_Dec2016/24.pdf.

Adams SM, Varaei S, Jalalinia F. Nurses’ Knowledge and Attitude towards Postoperative Pain Management in Ghana. Pain Res Manag. 2020. Aug 7;2020:4893707. doi: 10.1155/2020/4893707

Ung A, Salamonson Y, Hu W, Gallego G. Assessing knowledge, perceptions and attitudes to pain management among medical and nursing students: a review of the literature. Br J Pain. 2016;10(1):8–21. doi: 10.1177/2049463715583142

Greene, J.; El-Banna, M.M.; Briggs, L.A.; Park, J. Gender Differences in Nurse Practitioner Salaries. J. Am. Assoc. Nurse Pract. 2017, 29, 667–672, doi:10.1002/2327-6924.12512.

However, it is important to consider social gender differences as they may impact women's access to training. This is due to social priorities such as childcare and housework. The discussion is continued in the following response. References are attached:

Fabrizio S, Gomes DBP, Mendes Tavares M. COVID-19 She-Cession: The Employment Penalty of Taking Care of Young Children. WP/21/58. 2021. Available: https://www.imf.org/en/Publications/WP/Issues/2021/03/03/COVID-19-She-Cession-The-Employment-Penalty-of-Taking-Care-of-Young-Children-50117

A Badel A, Goyal R. When Will Global Gender Gaps Close?. WP/23/189. 2023. Available: https://www.imf.org/en/Publications/WP/Issues/2023/09/13/When-Will-Global-Gender-Gaps-Close-537981

Global Gender Gap Report 2023. World Economic Foro. 2023. Available: https://www3.weforum.org/docs/WEF_GGGR_2023.pdf

Mele BS., Holroyd-Leduc JM., Harasym P., Dumanski SM., Fiest K., Graham ID., et al. Healthcare workers’ perception of gender and work roles during the COVID-19 pandemic: a mixed-methods study. BMJ Open. 2021;11(12):e056434, doi: 10.1136/bmjopen-2021-056434.

4) In conclusion, I am confused as to whether to view the results of this study as a gender difference or an education difference.

We reviewed the introduction and discussion and clarified the issue of gender and education. In it, we discussed that the educational difference may be due to the gender gap that exists in all societies, even those with high economic income and full access to education. In these cases, the educational limitation is not related to a lack of resources, but to an aspect of the social context and the acquisition of certain responsibilities that are attributed to women, such as housework and childcare, and in many cases, they may put these before professional aspects, something that was more evident during the COVID-19 era.

References:

Fabrizio S, Gomes DBP, Mendes Tavares M. COVID-19 She-Cession: The Employment Penalty of Taking Care of Young Children. WP/21/58. 2021. Available: https://www.imf.org/en/Publications/WP/Issues/2021/03/03/COVID-19-She-Cession-The-Employment-Penalty-of-Taking-Care-of-Young-Children-50117

A Badel A, Goyal R. When Will Global Gender Gaps Close?. WP/23/189. 2023. Available: https://www.imf.org/en/Publications/WP/Issues/2023/09/13/When-Will-Global-Gender-Gaps-Close-537981

Global Gender Gap Report 2023. World Economic Foro. 2023. Available: https://www3.weforum.org/docs/WEF_GGGR_2023.pdf

Mele BS., Holroyd-Leduc JM., Harasym P., Dumanski SM., Fiest K., Graham ID., et al. Healthcare workers’ perception of gender and work roles during the COVID-19 pandemic: a mixed-methods study. BMJ Open. 2021;11(12):e056434, doi: 10.1136/bmjopen-2021-056434.

That's all for now. We wish you all the best in your revision work and look forward to publishing this manuscript as a research paper.

Yours sincerely, Reviewer

Reviewer 3 Report (New Reviewer)

Comments and Suggestions for Authors

The study uses multivariate logistic regression analysis to assess the impact of nurse gender on pain management practices. The study uses multivariate logistic regression analysis to assess the impact of nurse gender on pain management practices. However, the manuscript involved multiple comparisons, particularly in exploring various demographic and practice-related factors. This could inflate the risk of Type I error. I recommend applying a correction for multiple testing, such as the Bonferroni correction or the Benjamini-Hochberg procedure. The study's sample predominantly comprises female nurses, reflecting the general gender distribution in nursing. However, the authors might consider discussing whether this gender distribution in the sample, which is directly related to the context, impacts the generalizability of the results. The use of a self-administered questionnaire might introduce response bias. It would be beneficial if the authors could discuss how this potential bias was addressed or mitigated, as there is a subheading dedicated to mitigating bias. The study identifies gender-based differences in pain management practices. It would be valuable if the authors could elaborate on the potential practical implications of these findings, especially regarding training and protocol development in clinical settings.

Author Response

The study uses multivariate logistic regression analysis to assess the impact of nurse gender on pain management practices. The study uses multivariate logistic regression analysis to assess the impact of nurse gender on pain management practices. However, the manuscript involved multiple comparisons, particularly in exploring various demographic and practice-related factors. This could inflate the risk of Type I error. I recommend applying a correction for multiple testing, such as the Bonferroni correction or the Benjamini-Hochberg procedure.

The meaning of the reviewer's comment is unclear to us. We consulted a statistical expert in our institution to address the issues you raised. Below, we provide our response:

1.-Option 1: Our main variable for conducting contrasts is "gender". As stated in the "Variables" subsection of the manuscript, we initially asked participants in our questionnaire to choose between three possible categories for this variable: male, female, and non-binary. Unfortunately, we did not receive a response from a participant who identified as non-binary, so our variable automatically became dichotomous. We did not use any statistical tests in our analysis involving variables with 3 or more categories (ANOVA or Kruskal-Wallis) that required post hoc multiple comparison analysis. Therefore, we believe that no adjustment is necessary to minimize Type I error.

If you feel that we should perform further analysis, please do not hesitate to let us know.

2.- However, if the intention is to convey that the study includes numerous variables that could potentially impact the dependent variable, and as a result, some of the significant findings may be attributed to chance, it must be stated that this study does not establish a cause-and-effect relationship. Moreover, the variables analysed are more or less independent, as the questionnaire covers different concepts and is not a questionnaire with a conceptually homogeneous construct, or are 1000 genetic, biological, etc. markers.

Add that exact p-values of all comparisons have been described, allowing the reader to understand the impact on each variable. On the other hand, Bonferroni and Benjamini-Hochberg adjustments are commonly used in studies involving hundreds or millions of variables, such as gene studies. In this situation, there are few significant variables, and those that are significant may have clinical relevance that explains their significance. We believe that providing the study as is would help future research by providing reference variables.

Finally, we could opt not to assume differences above 5% (p-value <.05), but below 1% (<.01), but the variables that have had significance show relatively low p-values, many of them below .01. For this reason, it may not be necessary to make this correction. In spite of the above, if the reviewer considers it appropriate, we would be willing to make such a correction.

Summary of variables with significant p-values:

  • Perception of pain generated. p-value: .001
  • Specific ABG puncture training. p-value: .008*
  • Nurse perception of pain reduction from interventions:
    • Local anesthetic infiltration. p-value: .021*
  • Local anesthetic infiltration frequency. p-value: <.001**
  • Lack of knowledge of the anesthetic injection technique .011*
  • Lack of protocols or instructions in my unit. p-value: .008*

The study's sample predominantly comprises female nurses, reflecting the general gender distribution in nursing. However, the authors might consider discussing whether this gender distribution in the sample, which is directly related to the context, impacts the generalizability of the results.

Thank you for your comment. In fact, as we pointed out in the discussion of our article, the gender distribution of the participants is close to the real proportion of nurses published by official institutions. This aspect, added to the fact that the main objective of our study is exactly that, to find possible gender differences in pain management, means that the sample selected ensures the possible generalization of this objective. On the other hand, in the limitations we argue the overrepresentation of the female sex, but we insist that this is the guarantee of the population representativeness of nursing as a health profession in our environment.

Attached is a current reference justifying the existence of this inequality in the male nursing workforce:

Martsolf GR., Gigli K., Case B., Dill J., Dierkes A. Describing the male registered nursing workforce toward increasing male representation in professional nursing. Nursing Outlook. 2023;71(6):102081, doi: 10.1016/j.outlook.2023.102081.

The use of a self-administered questionnaire might introduce response bias. It would be beneficial if the authors could discuss how this potential bias was addressed or mitigated, as there is a subheading dedicated to mitigating bias.

Thank you for this suggestion. One intervention to avoid response bias was to make

several reminders and eventually a call for participation via social networks, so that

nurses who initially did not want to participate might eventually be motivated to

participate.

We have added this issue to the bias section.

The study identifies gender-based differences in pain management practices. It would be valuable if the authors could elaborate on the potential practical implications of these findings, especially regarding training and protocol development in clinical settings.

In this review we have expanded on the gender aspects of the gender gap in training. In addition, we have added a paragraph in the section 6 “Implications for practice” that elaborates on this issue.

Round 2

Reviewer 3 Report (New Reviewer)

Comments and Suggestions for Authors

my main comment was not addressed. It was aimed at ensuring the statistical integrity of the findings presented in the manuscript, particularly concerning the risk of Type I error that arises from conducting multiple comparisons. My recommendation to apply a correction for multiple testing stems from observing that, despite the main variable of interest being categorical (gender, with categories male and female or other), the manuscript engages in multiple comparisons across different demographic and practice-related factors as detailed in tables 1 and 2. The corrections should be considered in relation to the comparisons presented in Tables 1 and 2.

Author Response

Reviewer 3: my main comment was not addressed. It was aimed at ensuring the statistical integrity of the findings presented in the manuscript, particularly concerning the risk of Type I error that arises from conducting multiple comparisons. My recommendation to apply a correction for multiple testing stems from observing that, despite the main variable of interest being categorical (gender, with categories male and female or other), the manuscript engages in multiple comparisons across different demographic and practice-related factors as detailed in tables 1 and 2. The corrections should be considered in relation to the comparisons presented in Tables 1 and 2.

Authors: Thanks again for your comment. In this case, we have more clearly understood the change you requested in our manuscript.

For the variables included in Tables 1 and 2, we performed the Benjamini-Hochberg correction procedure as you suggested.

The adjusted p-values for Tables 1 and 2 are shown below:

Table 1. Characteristics of nurses and comparison of sociodemographic and laboral variables between male and female.

Total

p-value

Adjusted p-value

Use of pain interventions: YES

p-value

Adjusted p-value

Use of pain interventions: NO

p-value

Adjusted p-value

Age in years

.883

.942

.679

.933

.734

.897

Work experience as nurse in years

.942

.942

.773

.944

.731

.897

Nurse's workplace

.562

.876

.226

.640

.447

.897

Hospital

.281

.772

.060

.330

.684

.897

Type of employment contract

.708

.876

.565

.888

.939

.939

Table 2. ABG puncture related variables between male and female. 

Total

p-value

Adjusted p-value

Use of pain interventions: YES

p-value

Adjusted p-value

Use of pain interventions: NO

p-value

Adjusted p-value

Perception of pain generated

.001

.011

.021

.231

.029

.160

Puncture site

.717

.876

.469

.878

.816

.898

Specific ABG puncture training

.008

.044

.479

.878

.010

.110

ABG puncture / month

.632

.876

.967

.967

.402

.897

Skill perception performing arterial puncture

.087

.319

.233

.640

.358

.897

Allen test frequency

.714

.876

.928

.967

.588

.897

- So that the reader can see this adjustment, an extra column has been added next to the p-value in Tables 1 and 2, indicating that it is the p-value corrected by the Benjamini-Hochberg method.

- We have also included the performance of this adjustment in the statistical methods. We added the following:

“However, due to the multiple comparisons made with the different demographic and practice-related factors, all p-values were evaluated using the Benjamini-Hochberg procedure to control for false discovery rates.”

- We have reviewed the results section, being consistent with the new adjusted values.

- In addition, we have included the adjusted p-values in Figure S1 in the Supplementary Material.

Regarding this variable, which is presented in the supplementary material ("Perception of pain generated"), we would like to point out that for both subgroups the p-value exceeded the 0.05 threshold set for alpha after adjustment by the Benjamini-Hochberg procedure. However, if we visually analyse the box-and-whisker plot, we believe that there might be gender differences in those nurses who do not use any kind of pain management intervention. Although men and women share medians in this analysis, the distributions appear to be quite different. It is possible that with more men included in the study, this difference might be more evident.  We believe this is because your suggested adjustment for multiple comparisons is not entirely appropriate for our data set. It is difficult for our research team to understand a possible joint relationship between, for example, the hypothesized gender difference in the perception of pain generated, the frequency of use of the Allen test, the number of blood gases performed per month, and so on.

The variables included in our study are completely independent of each other, and the general construct of the questionnaire we used does not follow a homogeneous pattern. Furthermore, we believe that our research is based on the exploration of data and the discovery of new relationships, and that it is less critical to strictly control the FDR, especially since the results obtained can be used to generate hypotheses that will then be validated in future studies. This type of adjustment may be very useful in ecological studies, gene expression analyses, or studies where hundreds of biochemical variables may explain the presence of a disease. However, to the best of our knowledge, this adjustment does not seem to be common in studies similar to ours.

Thank you very much for your suggestions. We are at your disposal for any further clarification or additional information.

Round 3

Reviewer 3 Report (New Reviewer)

Comments and Suggestions for Authors

my concerns have been addressed

This manuscript is a resubmission of an earlier submission. The following is a list of the peer review reports and author responses from that submission.

Round 1

Reviewer 1 Report

Comments and Suggestions for Authors

Dear authors,

Thank you for the opportunity to review your manuscript. The article “Gender Differences Among Nurses in Managing Arterial Puncture-Related Pain: A Multicenter Cross-Sectional Study” is very interesting and well thought out. However, I have a few points of concern:

1.     The introduction section is really brief. You should add a few points of interest:

a.     I believe an introduction concerning gender differences in physiopathology is needed to introduce the topic (as an example you can refer to: Tokatli et al “Hormones and Sex-Specific Medicine in Human Physiopathology”).

b.     Spaces are missing before references (i.e: patients[3] should instead be patients [3].), this needs to be revised thorough the manuscript.

2.     In the methods:

a.     Study size should be corrected to “Sample size”.

b.     Significance value established for the p value should be reported in the methods.

3.     The results are very interesting and well written. A few minor points:

a.     In the tables, significant p values should be clearly indicated (with a * for example) to give the readers immediate visual feedback.

b.     In the results there are a few oversights, such as all approximation should be to the first decimal number, please revise.

4.     The discussion is engaging, but the comparison is almost completely missing:

a.     The first section is too repetitive of the results, please amend.

b.     A comparison with other studies investigating the issue should be reported, if available.

5.     Strengths and limitations, conclusions, as well as the rest of the manuscript are well-written and thought out.

Comments on the Quality of English Language

English is fine, just some oversights, no need for an English revision

Author Response

Thank you very much for your suggestions and contributions to our work. Please find attached the replies to your initial review.

Reviewer 2 Report

Comments and Suggestions for Authors

Dear authors,

This manuscript has the potential for publication. Please see my comments and suggestions below).

1 - Introduction

Page 2 - You should add this study's hypotheses and scientific questions. Please describe the motivation/justification of your research more clearly and finish it by highlighting the importance of your work.

2 - Materials and Methods

2.2.3. Study size (Line 83-85) - "As we mentioned before, this study is a subanalysis of a larger study, so for the sample size calculation, the prevalence of the use of interventions for the reduction of pain produced by ABG puncture was estimated generically". You could have provided more details on how you calculated the sample size. Alternatively, you could have developed a unique research and produced one complete paper. Another option would have been to show how you calculated the sample without citing the larger study. This would have provided more clarity and helped the reader better understand the research.

4 Results 

Page 6 - Table 2 - Nurse perception of pain reduction from interventions

If pain is a vital sign and a subjective experience to each person, how do you evaluate the response to the nurse about their perception of pain reduction from interventions in patients? The explanation of the topic in discussion is confusing because it is difficult to evaluate the perception of nursing about pain in other people.

Discussions

Page 9 (Line 239-245) - I suggest a new paragraph.

Page 9 and 10 (Line 253-273) - The paragraph is too extensive and discusses more than one topic. 

Page 10 (Line 284-285) - "We believe that the findings of this study may be generalizable to other nursing populations".

I suggest that you describe the area of study. In the method, you cite the setting, participants, and study size but do not cite the area of study coverage. Then, it would be best if you did not suggest that the findings of this study may be generalizable to other nursing populations. Nurses have different formations in other countries. Maybe it is necessary to develop other research in each region of different continents. I believe it is really important to do clinical research. Then, I suggest you refer to the importance of future study about the theme. 

Page 10 (line 294-296) - "In our opinion, the unequal application of pain relief strategies according to the sex of the health care provider has no justification and can be classified as gender bias"

The paper is not an opinion paper, and you do not have enough data to determine gender bias.

Author Response

(The authors gave the same response as above.)

Reviewer 3 Report

Comments and Suggestions for Authors

Dear authors, thank you for allowing me to review this interesting manuscript focused on assessing gender differences in nurses' management of arterial punctures.

I found the manuscript well-written, with a well-explained rationale and a considerable sample size. Moreover, I think the topic presented is of interest for nurses and healthcare professionals, as they are the most frequently involved in ABG procedures.

Although I consider this manuscript worth consideration for the international readership, I have found some points of concern that should be considered before I could recommend it for publication. I list above the main points of concern I have identified.

Introduction. It is well organized and takes the reader straight to the point (research hypothesis).

Page 1, lines 39–41. Please explain better what you intend with "gender bias", as I found this concept to be too broad to be well intended by someone who does not have a clear underpinning for this concept.

Page 1, lines 41–44 Please provide at least a reference for what you have said about the prevalence of nurses performing the ABG sampling and the fact that the majority of RN in your setting are female.

Page 2, lines 45–46 Please explain better how pain has been analyzed and what results in particular were found.

Methods.

I have not found in any part how many people have replied to the questionnaire out of the total number of questionnaires that were sent (response rate). You have provided a sample size of 528 participants, which is probably the same number of participants you have included, but not the denominator to which this number should refer.

The rest of the section is good, and I have appreciated the description you have provided about the instruments and the data analysis.

The results are well organized and presented, and I have no suggestions to provide regarding this part.

The discussion is very poorly sustained by the literature and repeats almost the same results that were presented before. I am talking about page 9, lines 221-224, and lines 237-251. These are repetitions of the results and are not useful to the readers. What could be performed is a fair discussion of the obtained findings against the literature, rather than repeating the data that were provided in the previous section. And even when an explanation has been provided (see pages 9–10, lines 269–270), it seems to be uncomprehensible.

Conclusions. Need to be rewritten, not only emphasizing the obtained results, but also thinking about what nursing/hospital managers or decision makers should do to reply to the identified problem. Moreover, an implication for research is needed to give the manuscript the quality that should be reached.

I hope my suggestions help you in improving the quality of your manuscript

Author Response

(The authors gave the same response as above.)

Round 2

Reviewer 3 Report

Comments and Suggestions for Authors

Dear authors, thank you for allowing me to review this second version of your manuscript. I found it considerably improved and I am very satisfied for the consideration you have showed on all my previous comments. Thus, I congratulate you and recommend this manuscript for publication.